# Comparison of Spectral-Domain OCT versus Swept-Source OCT for the Detection of Deep Optic Disc Drusen

**DOI:** 10.3390/diagnostics12102515

**Published:** 2022-10-17

**Authors:** Simon P. Rothenbuehler, Lasse Malmqvist, Mohamed Belmouhand, Jakob Bjerager, Peter M. Maloca, Michael Larsen, Steffen Hamann

**Affiliations:** 1Department of Ophthalmology, Rigshospitalet, University of Copenhagen, 2600 Glostrup, Denmark; 2Department of Ophthalmology, OCTlab, University Hospital Basel, 4031 Basel, Switzerland; 3Institute of Molecular and Clinical Ophthalmology Basel (IOB), 4031 Basel, Switzerland; 4Moorfields Eye Hospital NHS Foundation Trust, London EC1V 2PD, UK; 5Faculty of Health and Medical Sciences, University of Copenhagen, 2200 Copenhagen, Denmark

**Keywords:** optical coherence tomography, optic disc drusen, optic nerve head drusen, swept-source, spectral-domain, enhanced depth imaging, comparison

## Abstract

Deep optic disc drusen (ODD) are located below Bruch’s membrane opening (BMO) and may go undetected due to the challenges in imaging them. The purpose of this study is a head-to-head comparison of currently widely used imaging technologies: swept-source optical coherence tomography (SS-OCT; DRI OCT-1 Triton, Topcon) and enhanced depth imaging spectral-domain optical coherence tomography (EDI SD-OCT; Spectralis OCT, Heidelberg Engineering) for the detection of deep ODD and associated imaging features. The eyes included in this study had undergone high-resolution imaging via both EDI SD-OCT and SS-OCT volume scans, which showed at least one deep ODD or a hyperreflective line (HL). Grading was performed by three graders in a masked fashion. The study findings are based on 46 B-scan stacks of 23 eyes including a total of 7981 scans. For scan images with ODD located above or below the level of BMO, no significant difference was found between the two modalities compared in this study. However, for HLs and other features, EDI SD-OCT scan images had better visualization and less artifacts. Although SS-OCT offers deep tissue visualization, it did not appear to offer any advantage in ODD detection over a dense volume scan via EDI SD-OCT with B-scan averaging.

## 1. Introduction

Optic disc drusen (ODD) are noncellular calcified deposits within the optic nerve head (ONH) of the human eye, with a prevalence of approximately 2% in the general population [1]. When located deep within the ONH, ODD can cause optic disc elevation. The appearance on fundus examination can mimic optic disc edema and prompt urgent and invasive diagnostic procedures to exclude increased intracranial pressure and other serious neurological disease. In short, deep ODD can pose a significant diagnostic challenge with clinical implications [2]. Over time, ODD show an association with retinal nerve fiber layer loss and corresponding visual field defects [3,4], which are present in up to 87% of patients [5]. Furthermore, ODD are detected in association with vascular disease such as non-arteritic anterior ischemic optic neuropathy, central retinal artery occlusion, central retinal vein occlusion, and peripapillary choroidal neovascularization [6,7].

ODD can be of variable size and number; in most cases they are bilateral. ODD may be readily recognized during fundoscopy by their characteristic protuberant yellowish appearance. In other instances, ODD may be invisible during a fundoscopy; these are the so-called buried ODD, which can only be detected via other diagnostic modalities. Especially when singular and small, ODD can easily be missed and are presumably underdiagnosed in many clinical settings.

The diagnosis of ODD has been enhanced by the advances in noninvasive ophthalmic imaging from ophthalmic ultrasonography to fundus autofluorescence to optical coherence tomography (OCT) and the detection rate of even the smallest ODD has been steadily increasing [8,9,10]. The development of enhanced depth imaging (EDI) for spectral-domain OCT (SD-OCT) by Spaide et al. [11] has further improved the visibility of structures deep within the ONH, such as the lamina cribrosa (LC), ODD, and ODD-associated features. Currently, a dense high-resolution EDI SD-OCT volume scan of the ONH is the recommendation by the Optic Disc Drusen Studies Consortium for ODD diagnosis [12]. The technology of swept-source OCT has been developed to achieve high scan rates and deep tissue visualization based on longer wavelength laser light. More recently, OCT angiography (OCTA) was introduced for the simultaneous noninvasive visualization of vascular flow and structural information. This modality is now available in both SD-OCT and SS-OCT devices and can facilitate the detection of ODD-associated changes such as retinal vein occlusion or arterial occlusion, vascular anomalies, and peripapillary choroidal neovascularization [13,14]. Furthermore, OCTA is investigated as a biomarker predicting visual loss due to ODD [15].

While a dedicated EDI SD-OCT scan is the current recommendation for ODD detection, a fast and comprehensive ONH scan protocol would be helpful in routine clinical settings. For a one-fits-all approach, a single dense SS-OCT volume scan with OCTA could potentially deliver the visualization of relevant ONH structures and vascular flow at a penetration depth and resolution suitable for ODD detection.

In this retrospective study, a set of experienced graders compared EDI SD-OCT against SS-OCT with OCTA in eyes with deep ODD or ODD-associated changes, e.g., hyperreflective lines (HLs) located below the level of Bruch’s membrane opening (BMO).

With increasing tissue depth—and sometimes decreasing size—deep ODD are one of the most demanding diagnostic challenges. A set of eyes with deep ODD is ideal for putting the current imaging modalities to the test, for comparing OCT technologies, and for potentially guiding future imaging recommendations for ODD diagnosis.

## 2. Materials and Methods

We performed a retrospective comparative study of SD-OCT and SS-OCT imaging modalities for detecting ODD and associated changes located deep in the ONH. Included in this study were patients who had undergone SD-OCT EDI imaging performed according to the Optic Disc Drusen Studies Consortium recommendations [12], as well as a high-resolution SS-OCT angiography scan of the ONH as part of standard clinical care. Other inclusion criteria were the presence of at least one ODD or a part of one ODD below the level of BMO, i.e., a deep ODD, (Figure 1), and/or the presence of prelaminar HLs below the level of BMO. Furthermore, the presence of additional superficial ODD, i.e., ODD located above the level of BMO, were allowed if the changes were not extensive enough to visibly obstruct the imaging of deeper ONH structures. Exclusion criteria were an image quality score lower than 20 and 40 for SD-OCT and SS-OCT, respectively, or scan properties that differ from the settings mentioned subsequently. Patients were imaged between November 2018 and November 2020 at the Department of Ophthalmology, Rigshospitalet, Glostrup, Denmark, when they visited the neuro-ophthalmology clinics for ODD workups. We adhered to the Declaration of Helsinki in conducting this study.

A Spectralis OCT device from Heidelberg Engineering, Heidelberg, Germany, was used for SD-OCT imaging. The imaging settings included a dense horizontal scan pattern of 15 × 10°, manually centered on the ONH, with 97 b-scans. EDI mode and high-resolution mode were enabled, with active eye tracking and automatic scan averaging set to 30. A DRI OCT-1 Triton device from Topcon, Tokyo, Japan, was used for the SS-OCT imaging. A high-resolution 6 × 6 mm OCTA volume scan centered on the ONH was acquired, with eye tracking enabled, yielding 512 B-scans. Scan averaging was unavailable. Both scan modalities were acquired after pupil dilation on the same day by experienced operators.

For both devices, the B-scans were exported as a stack of uncompressed bitmap format images and then cropped to omit extraneous information such as scale bars or accompanying fundus imagery (see Figure 2). There was no indication or labeling on the images identifying the OCT technology to the graders. For SD-OCT, all 97 B-scans were included in the grading. For SS-OCT, a consecutive set of 250 B-scans of the 512 B-scans performed were selected to match the corresponding scan area of the 97 SD-OCT B-scans. Only structural B-scans were assessed, while OCTA imaging data was not part of this comparative analysis. Each image stack was assigned a random identification number and loaded into the ImageJ software application (ImageJ version 2.0.0-rc-68/1.52 h) for grading. As a grading reference within the stack, consecutive B-scan numbers were added to the images as labels. No contrast enhancement or other additional image processing was performed.

The image stacks were graded in random order by three experienced graders (S.H., L.M., and S.P.R.). Similar to the procedure in a clinical setting, this allowed the graders a stack-wise assessment, with fluid back and forth scrolling through the images. For differentiation of ODD, neighboring B-scans were assessed for continuity of imaging features such as vessels and other ONH structures.

The following imaging features were graded based on the number of positive scans within a stack: ODD above the level of BMO, ODD below the level of BMO, HL above the level of BMO, HL below the level of BMO, and the presence of peripapillary hyperreflective ovoid mass-like structures (PHOMS). PHOMS are located superficially around the ONH in an arcuate or ring-like configuration, and are believed to represent axonal stasis [16]. HLs are often observed in cases with ODD and may represent precursors to ODD in these cases [17].

For the aforementioned features, the sequences of positive scans (e.g., scan number 25 to 30) were recorded on an Excel sheet (Microsoft Excel for Mac, version 16.26) and the ratio of the positive scans to the total number of scans in the stack was calculated. The average ratio across all three graders was recorded for analysis.

Furthermore, the following features were graded on the basis of having or not having at least one occurrence within the entire stack: LC level visibility (LCLV) and presence of motion artifacts. For categorical values, the statistical mode of the grade given by the three graders was selected as the final result, which was used in the analysis.

The Chi-square test or Fisher’s exact tests were used for the comparison of categorical data and the Student’s *t*-test was used for comparison of continuous data. *p*-values of ≤0.05 were treated as statistically significant. The statistical analyses were performed using SAS Studio 3.8 (SAS Institute, Cary, NC, USA).

## 3. Results

Thirty-one eyes of 17 patients, each with ODD and/or HLs located deep within the ONH, were included in this retrospective comparative non-interventional case-series. Eight eyes were excluded because of insufficient scan quality or diverging scan settings (e.g., lower B-scan number or low resolution) for either modality. For the SD and SS modalities, a combined number of 46 stacks, with a total of 7981 B-scans, were exported for grading.

Of the total 7981 B-scans, the average of the three graders’ scoring was as follows: 914.7 (11%) B-scans positive for superficial ODD, 1005.7 (13%) positive for deep ODD, 345.7 (4%) positive for superficial HL, 842.3 (11%) positive for deep HL, and 1573.7 (20%) positive for PHOMS. The reported grading results per study eye are shown in Table 1.

For scans with superficial ODD, there was no difference in the ratio of positive scans found per stack for SD-OCT (mean = 0.12) and SS-OCT (mean = 0.11; *p* = 0.71). Similarly, there was no difference between the SD-OCT scans (mean = 0.12) and the SS-OCT scans (mean = 0.13) for deep ODD (*p* = 0.91). For HLs located above the level of BMO, there was a significantly higher ratio per stack among the SD-OCT scans (mean = 0.11) than among the SS-OCT scans (mean = 0.02; *p* = 0.047). Furthermore, HL located below the level of BMO were found in a significantly higher ratio per stack for SD-OCT (mean = 0.23) than for SS-OCT scans (mean = 0.06; *p* = 0.016).

For PHOMS, a significantly higher ratio of scans per stack was positive for SD-OCT (mean = 0.25) than for SS-OCT (mean = 0.18; *p* = 0.016).

While assessing the LC level within a stack, SD-OCT yielded significantly better visibility than SS-OCT (*p* = 0.0003). The presence of visible motion artifacts was significantly higher in the SS-OCT stacks than in the SD-OCT stacks (*p* = 0.049). Lastly, all three graders subjectively noted an overall higher level of contrast for the SD-OCT scans than for the SS-OCT scans.

## 4. Discussion

This SS-OCT versus SD-OCT study was performed to compare the currently widely used OCT technologies head-to-head in terms of their performance visualizing deep ODD and associated imaging features such as HLs, PHOMS, and LC level. The high-resolution SD-OCT volume scan with EDI mode and scan averaging enabled is the diagnostic method of choice recommended by the Optic Disc Drusen Studies Consortium [12].

Deep ODD are conventionally located below the level of BMO and within the ONH space, which is confined by the scleral canal walls and the LC. Small and singular deep ODD commonly go undetected when only fundoscopy is employed. They, therefore, represent the presumably underdiagnosed spectrum of ODD for which high ophthalmic imaging performance is crucial. The recommended and dedicated EDI SD-OCT scan protocol facilitates detection of the smallest and most deeply located ODD. However, it can be time consuming and its purposeful use tends to be limited to specialized clinics.

A more versatile scan protocol could yield information on multiple aspects of the ONH, such as structural information (on the retinal nerve fiber layer, ganglion cell layer, vitreous body status), vascular flow (via OCTA), and anatomical landmarks (e.g., LC, BMO), in addition to ODD detection, all in one go. Such an all-in-one scan approach would be easily integrated into daily practice and more widely used. Therefore, it is necessary to compare new imaging technology to the currently established recommendations and to attempt to refine scan protocols to promote wider use.

Our findings show comparable ODD detection rates for SS-OCT and EDI SD-OCT. There was no significant difference in the detection of ODD located above the level of BMO (i.e., superficial ODD). Similarly, and notably, there is no difference in the detection of ODD below the level of BMO (i.e., deep ODD) between the two OCT technologies.

For ODD-associated changes such as HLs, which are also categorized by the location above or below BMO, our results showed significantly better visualization delivered by EDI SD-OCT than by SS-OCT. Similarly, PHOMS are significantly better visualized by SD-OCT than SS-OCT. Furthermore, SD-OCT delivered superior visualization of the LC level in the stack overall. In this study, some SS-OCT stacks showed visible motion artifacts, while SD-OCT scans showed none (see Table 1).

The study findings indicate comparable SS-OCT and EDI SD-OCT capability to visualize ODD within the ONH. SS-OCT technology, with its longer wavelength light, shows no advantage over EDI SD-OCT in this setting. Regarding HLs, LCLV, and PHOMS visibility, EDI SD-OCT delivered better visualization than SS-OCT.

We assume that these differences are largely attributable to the B-scan averaging method in combination with active eye tracking in the SD-OCT device. The robust eye-tracking mechanisms in the SD-OCT device almost completely compensated for or discarded B-scans affected by motion artifacts during acquisition. Eye tracking is also available in the SS-OCT device but is apparently less effective. A post hoc analysis via an enface projection of the B-scan stacks revealed small misalignments in most of the SS-OCT scans that were presumably due to eye movements. In many cases these changes were too small to be reported as a motion artifact by the graders but might nevertheless have affected image quality.

Regarding the ONH landmark with the deepest location (i.e., LCLV), the SD-OCT device again outperformed, most likely because of the scan averaging feature. Averaging was set to 30 scans per line, which, by visibly reducing speckle noise, also contributed to the high contrast reported by the graders.

Although SS-OCT devices offer scan averaging in single-line mode and some other scan patterns, it is not available for the acquisition of volume scans. For reasons that we could not establish, SS-OCT devices from other manufacturers do not offer B-scan averaging for volume scans. This is notwithstanding the fact that an OCTA algorithm inherently has to receive multiple scan passes of a single location to calculate flow.

Based on the findings of this comparative study, we speculate that SS-OCT, in combination with higher number scan averaging, could further increase the depth and contrast of tissue visualization.

The strengths of our study include the overall high B-scan numbers graded stack-wise and similar to the clinical setting. To the best of our knowledge this is the first report on a comparison of SS-OCT and SD-OCT for ODD detection.

A weakness of this study is the limited number of eyes included yet a high number of scans with the entailing workload through manual grading. Furthermore, because of differing B-scan spacing in the two devices, our setup did not allow for a scan-by-scan analysis to compare single lesions.

## 5. Conclusions

In conclusion, high-resolution SD-OCT imaging combined with EDI remains the recommended OCT modality for ODD detection (or exclusion) and precise follow-up. For ODD screening in low suspicion cases, an SS-OCT volume scan of the ONH could yield a reasonable trade-off with high versatility and speed. Generally, high signal penetration depth is needed—ideally down to the LC level—to visualize deep ONH structures reliably. B-scan averaging and the performance of eye tracking seem to be important factors in deep tissue visualization. Ultimately, it can be expected that the incidental detection rate of small and deep ODD will increase and that the clinical implications regarding visual field loss and associated vascular disease need to be further investigated.

As a rule of thumb, if there is LCLV on a meaningful portion of the B-scans within a stack, visualization or exclusion of even small and deep ODD can be assumed. Future research is needed to compare new imaging methods and to define a current gold standard method.

## Figures and Tables

**Figure 1 diagnostics-12-02515-f001:**
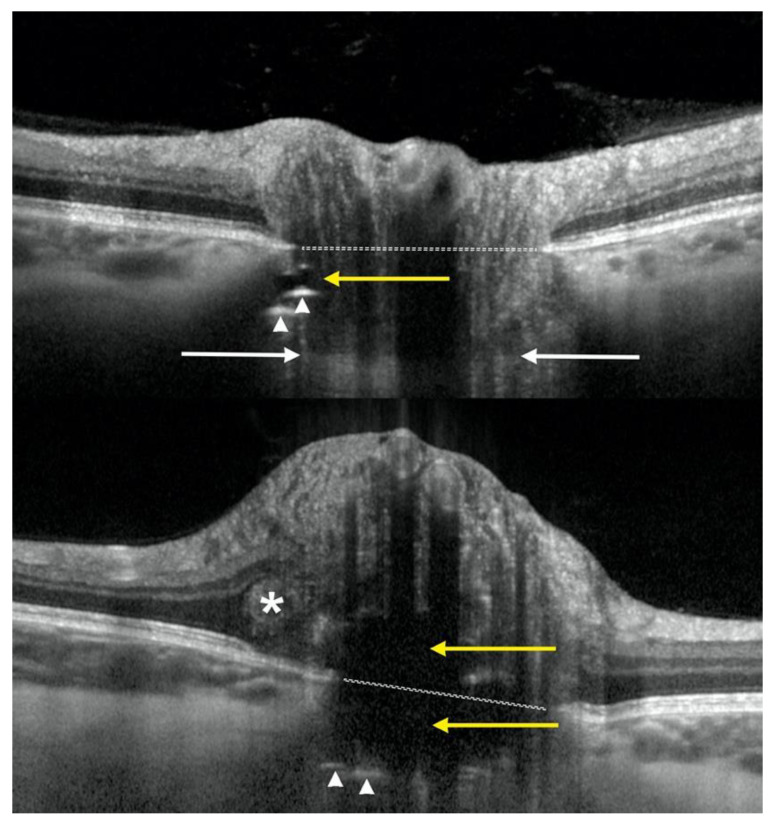
Enhanced depth imaging OCT showing two cases of deep optic disc drusen with associated features and landmarks: level of Bruch’s membrane opening (dotted white line) and visible level of lamina cribrosa (white arrows), optic disc drusen as signal-poor, hyporeflective irregular shaped areas (yellow arrows) with slightly hyperreflective margins, prelaminar hyperreflective lines (white arrow heads), and PHOMS (asterisk).

**Figure 2 diagnostics-12-02515-f002:**
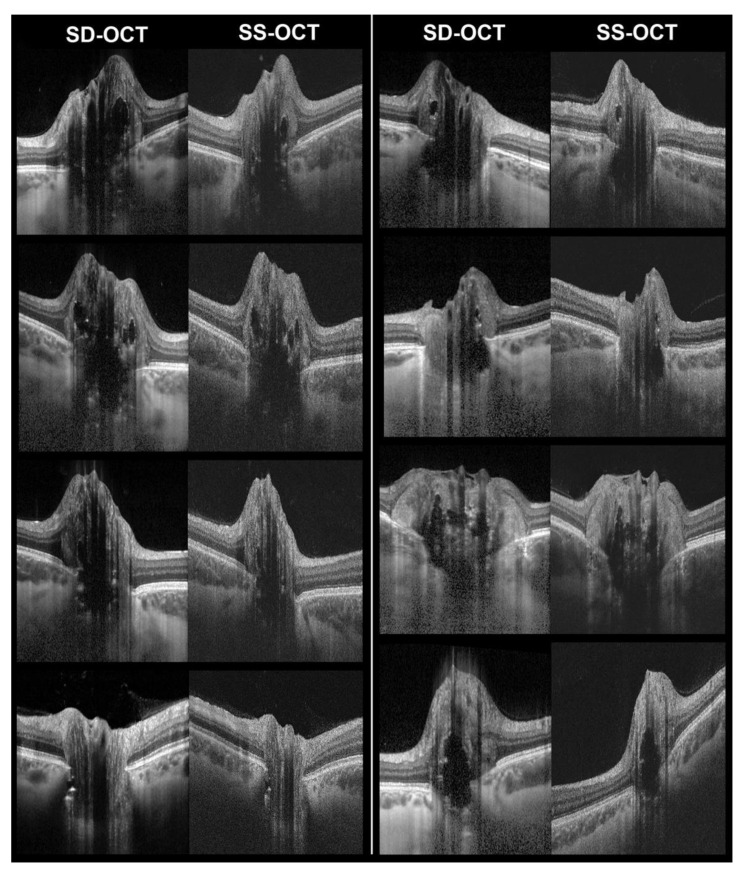
Examples of corresponding B-scans with enhanced depth imaging spectral-domain OCT (SD-OCT) and swept-source OCT (SS-OCT). Images show optic disc drusen (hyporeflective bodies with hyperreflective margins) and some associated features, such as hyperreflective lines and PHOMS.

**Table 1 diagnostics-12-02515-t001:** Color visualization of the grading results for swept-source (SS) OCT and spectral-domain (SD) OCT: % of positive scans (average across graders) or 1/0 for categorical data (statistical mode of grader’ responses) in a continuous red (=100% or 1) to green (=0% or 0) color scale. Graded imaging features: lamina cribrosa level visibility (LCLV), presence of motion artifacts (MA), ODD above the level of BMO (superficial ODD), ODD below the level of BMO (deep ODD), hyperreflective lines above the level of BMO (superficial HL), HL below the level of BMO (deep HL), and PHOMS.

EyeID	LCLV			MA			Superficial ODD			Deep ODD			Superficial HL			Deep HL			PHOMS	
	SD	SS		SD	SS		SD	SS		SD	SS		SD	SS		SD	SS		SD	SS
01	1	0		0	0		0.0%	0.0%		10.7%	10.0%		0.0%	0.0%		19.9%	4.5%		0.0%	0.0%
02	1	1		0	0		5.8%	0.0%		17.2%	17.7%		0.0%	0.0%		27.5%	2.5%		0.0%	0.0%
03	1	0		0	0		1.0%	1.1%		5.2%	3.5%		0.0%	0.0%		4.1%	1.3%		0.0%	0.0%
04	1	0		0	0		0.0%	0.0%		0.0%	0.0%		0.0%	0.0%		11.3%	5.3%		0.0%	0.0%
05	1	1		0	0		0.0%	0.0%		1.4%	0.9%		0.0%	0.0%		16.8%	4.8%		0.0%	0.0%
06	1	1		0	0		0.0%	0.0%		7.9%	4.3%		0.0%	0.0%		16.2%	6.3%		0.0%	0.0%
07	1	1		0	0		0.0%	0.0%		11.7%	6.5%		0.0%	0.0%		8.2%	1.2%		0.0%	0.0%
08	1	1		0	0		0.0%	1.5%		12.0%	8.7%		0.0%	0.0%		28.2%	17.1%		0.0%	0.0%
09	1	1		0	0		0.0%	0.0%		5.8%	0.1%		0.0%	0.0%		11.0%	1.9%		0.0%	0.0%
10	1	0		0	0		0.0%	0.0%		2.4%	0.0%		0.0%	0.0%		4.5%	0.0%		0.0%	0.0%
11	1	0		0	0		37.5%	16.0%		34.7%	18.4%		1.7%	1.9%		17.5%	0.0%		56.0%	22.9%
12	1	0		0	1		0.0%	0.0%		13.1%	10.9%		0.0%	0.0%		40.2%	13.5%		0.0%	19.1%
13	1	0		0	0		0.0%	0.0%		12.0%	11.1%		0.0%	0.0%		14.1%	2.8%		1.7%	0.0%
14	1	0		0	0		0.0%	0.0%		12.4%	9.1%		0.0%	0.0%		4.5%	0.0%		3.1%	0.0%
15	1	0		0	1		0.0%	0.0%		9.6%	9.1%		0.0%	3.5%		22.3%	6.1%		1.7%	0.0%
16	0	0		0	0		19.2%	13.3%		6.5%	13.2%		47.8%	12.0%		50.9%	25.1%		80.8%	43.6%
17	0	0		0	0		72.2%	77.3%		21.6%	32.9%		73.5%	0.0%		23.4%	11.2%		100.0%	100.0%
18	1	0		0	0		30.2%	43.3%		19.9%	57.2%		42.3%	6.9%		52.9%	0.0%		66.0%	53.9%
19	1	0		0	0		23.0%	37.1%		40.9%	41.9%		6.9%	5.3%		46.4%	23.1%		58.8%	51.6%
20	0	0		0	1		65.6%	49.7%		0.0%	3.6%		62.9%	14.1%		23.4%	3.3%		100.0%	78.4%
21	0	0		0	1		0.0%	0.0%		0.0%	0.0%		0.0%	0.0%		31.6%	0.9%		22.7%	0.0%
22	1	0		0	1		0.0%	0.0%		4.5%	3.2%		1.0%	0.0%		14.4%	3.3%		17.9%	0.0%
23	1	0		0	0		18.6%	20.5%		36.8%	28.9%		7.6%	0.0%		33.0%	0.0%		68.7%	36.0%

## Data Availability

The data that support the findings of this study are available from the corresponding author, SPR, upon reasonable request.

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
