# Peer review of "Comparison of Spectral-Domain OCT versus Swept-Source OCT for the Detection of Deep Optic Disc Drusen"

_diagnostics, 2022, doi:10.3390/diagnostics12102515_

Round 1
Reviewer 1 Report
Abstract: I would consider removing diagnostic challenge to actually stating that they may go undetected. This is clinically important for NAION.
Abstract: I would like to see the names of the OCT platforms up front and central in the abstract
Abstract: consider the word masked versus blinded
Abstract (and later in the conclusions), you have small numbers. I would suggest to tone it down and say something like: The Triton SS-OCT does not appear to offer any advantage over HE SD-OCT in detection of ODDs.
It is important to put the platforms as in the future SS-OCT on another platform may be less automated and tracking may be better ...and actually proved to be the superior modality.
Introduction: Good summary.
Methods: Good, I would be able to take the methods and apply them to reproduce the study.
Methods: for the figure could you split this into two? Really to increase the magnification of the images. This is not a deal breaker if you cannot.
Results
The first two sentences are a bit clunky. I would remove:
....in this retrospective study.
Remove the word
Prior
Can we make sure table 1 is cited in the results and not in the discussion section of the article? Please move the Table 1 results into results with those comments line 199 onward.
Table 1...could you add a colour scale; remove the number of scans, as you have already talk about this. I would like to see some more discussion about this as although it is not statistically significant, there was some instances of large disagreement e.g. eye number 11. This would suggest to me that you probably need more power to be really certain of your conclusions.
While you have commented that a strength of the study is the grading, you have not provided the agreement amongst the graders. Therefore I would not consider this a demonstrated strength. I believe it would be the first reported analysis of SD and SS in ODDs?
Likewise with the limitations of the study I would suggest that you attempt to mitigate this statement with a sentence about the manual workload and a future study could include a machine learning tool to reduce workload and increase the number of patients/eyes.
I would add a future research statement (two sentences).
I would not conclude that SD remains the gold standard, as this is a small study. I would just state the findings stating with For ODD screening.
Author Response
Reviewer 1
We thank the editor and the reviewers for the time and effort spent to improve our manuscript. We have carefully read their comments and worked on the manuscript to incorporate and reflect their valuable suggestions.
Below we provide a point-by-point response to the reviewer’s inputs:
- The abstract has been reformulated to include that deep ODD may go undetected due to the challenges in imaging them.
- Furthermore, it has been amended to include the device names (DRI OCT-1 Triton, Topcon and Spectralis OCT, Heidelberg Engineering) and the word “blinded” has been replaced by “masked".
- The abstract conclusion has been rephrased to “…SS-OCT did not appear to offer any advantages in ODD detection…”
- Figure 1 and 2 have been enlarged to increase visibility of the OCTs and clarify the notations.
- The result section has been rephrased to include the more accurate information of a “retrospective, comparative, non-interventional case-series” to follow the suggestion of another reviewer. The sentences have been rephrased and the word prior has been omitted.
- Reference to Table 1 has been moved to the results section together with statements regarding lamina cribrosa visibility and motion artifacts. Comments regarding these features were rephrased and expanded in the discussion section. Furthermore, scan numbers were deleted from table 1. Regarding the color scale (green to red) we see no way of improving the visualization, assuming the table can be reproduced shown via screenshot in the manuscript. Please let us know if this reviewer comment has been misunderstood.
- Disagreement between the two modalities or platforms were indeed the result of this comparison. SS-OCT contrast and visualization of the imaging features was often weaker as subjectively noted by the graders. The graders only had access to b-scan images, and only larger “jumps/glitches” within a stack were reported as motion artifacts. Nevertheless, a review of the original data in an enface projection revealed minor artifacts or misalignments (presumably due to motion) affecting most of the SS-OCT scans. Presumably this is inherent to the eye tracking used and device specific processing. In the case of eye 011, the scans contained some of these minor artifacts where deep ODD occurred compounded with PHOMS, some disc edema and dense vessel structures. Presumably this led to the larger difference between SD and SS-OCT in all categories as table 1 indicates. The discussion was amended to include this information.
- Throughout the manuscript the wording “gold standard” was removed in connection with OCT as the literature still holds ultrasound as the gold standard method, as it was rightly pointed out by another reviewer.
- A future research statement was added calling for investigation of new imaging platforms and definition of a gold standard method.
Reviewer 2 Report
I had great pleasure to read the article by Rothenbuechler et al.
Optic disc drusen (ODD) are proteinaceous materials made of calcium, mucopolysaccharides and amino nucleic acid deposits . They can be superficially located in the papilla (superficial drusen) or even more deeply (deep drusen, also known as “buried drusen”), thus altering the optic disc contour. . Currently, the gold standard in the ODD diagnosis is represented by US, which shows hyperreflective lesions at the optic disc level, followed by acoustic shadowing (J. Clin. Med. 2019, 8, 1449). Moreover in a recent paper (J. Clin. Med. 2022, 11, 3715) 20% of ODD were not detected by enhanced depth imaging (EDI) SD-OCT
Some minor points for the present version of the manuscript should be clarified.
The authors utilized echography to make diagnosis of ODD?
Page 3 line 109 .Why the authors exported Bscan images as bitmap format?
Page 5 line 138 please specify version of Excel.
Reference list should be improved (see above)
Author Response
Reviewer 2
We thank the reviewers for the time and effort spent to improve our manuscript. We have carefully read their comments and worked on the manuscript to incorporate and reflect their valuable suggestions.
Below we provide a point-by-point response to the reviewer’s inputs:
- As suggested by the reviewer we have read the two studies regarding the current gold standard method for ODD diagnosis: Rajagopal et al. "Detection of Optic Disc Drusen in Children Using Ultrasound through the Lens and Avoiding the Lens—Point of Care Ultrasound Technique of Evaluation Revisited." Journal of Clinical Medicine 8.9 (2019): 1449; and Rosa et al. "Optic Nerve Drusen Evaluation: A Comparison between Ultrasound and OCT." Journal of Clinical Medicine 11.13 (2022): 3715.
The study by Rajagopal et al. compares two methods of ocular ultrasound for ODD diagnosis (with and without lens avoidance) but not using OCT. US is described as the gold standard method citing a study of Auw-Haedrich et al. from 2002, well before the first description of EDI-OCT by Spaide et al. in 2008. We agree that the literature still holds US as the gold standard method although it has been repeatedly put in question by studies using various advances of OCT over the past two decades. We therefore revised the manuscript to state OCT only as the ODDS Consortium recommended method.
The more recent study by Rosa et al. compared US and EDI-SD-OCT for ODD detection claiming to apply ODDS Consortium recommendations for the OCT scans. However, the description of the scan settings and the figures provided in the publication are inconsistent and point to a much lower spatial resolution (scan area 8.9x8.9mm with 73 lines equals a b-scan spacing of approx. 121µm vs the recommended 30µm). Furthermore, the authors do not state whether scan averaging was used and at what rate (recommended is 30 or more). As our results point out, this is a crucial aspect of the recommended imaging method - especially regarding the detection of deep ODD. In our opinion the results of the study by Rosa et al. are not based on the recommendations and not representative. We therefore refrain from including the study in the references of our manuscript. Nevertheless, as stated above, we revised the manuscript regarding the gold standard wording. - In our study the diagnosis of ODD was made solely by recommended OCT imaging.
- The software of both devices offered bitmap as an uncompressed image file export option. An uncompressed image file format was selected to avoid image degradation by compression algorithms such as JPEG.
- The software version of Microsoft Excel was added to the manuscript.
Reviewer 3 Report
The Manuscript id well written and the study is well designed ; minor changes in the study design is to add "retrospective, comparative , non interventional, Case series of 13 patients of 23 eyes,,,,,
Author Response
Reviewer 3
We thank the reviewers for the time and effort spent to improve our manuscript. We have carefully read their comments and worked on the manuscript to incorporate and reflect their valuable suggestions.
Below we provide a point-by-point response to the reviewer’s inputs:
- The result section has been rephrased to include the more accurate description of a “retrospective, comparative, non-interventional case-series.” Following the suggestions of another reviewer the beginning of the paragraph has been rephrased and the word “prior” has been omitted.